# Thiol-ene Reaction: An Efficient Tool to Design Lipophilic Polyphosphoesters for Drug Delivery Systems

**DOI:** 10.3390/molecules26061750

**Published:** 2021-03-20

**Authors:** Stéphanie Vanslambrouck, Raphaël Riva, Bernard Ucakar, Véronique Préat, Mick Gagliardi, Daniel G. M. Molin, Philippe Lecomte, Christine Jérôme

**Affiliations:** 1Center for Education and Research on Macromolecules (CERM), CESAM Research-Unit, University of Liège, Allée du 6 août 13, B6a, Sart-Tilman, 4000 Liège, Belgium; stephanie.vsb@outlook.com (S.V.); raphael.riva@uliege.be (R.R.); Philippe.Lecomte@uliege.be (P.L.); 2Advanced Drug Delivery and Biomaterials, Louvain Drug Research Institute, Université Catholique de Louvain, Avenue Mounier, 73, B1.73.12, 1200 Brussels, Belgium; bernard.ucakar@uclouvain.be (B.U.); veronique.preat@uclouvain.be (V.P.); 3Department of Physiology, Faculty of Health, Medicine and Life Science, Maastricht University, Universiteitssingel 50, 6229 ER Maastricht, The Netherlands; m.gagliari@maastrichtuniversity.nl (M.G.); d.molin@maastrichtuniversity.nl (D.G.M.M.)

**Keywords:** click chemistry, biodegradable polymer, polyphosphoester, drug-delivery

## Abstract

Poly(ethylene glycol)-*b*-polyphosphoester (PEG-*b*-PPE) block copolymer nanoparticles are promising carriers for poorly water soluble drugs. To enhance the drug loading capacity and efficiency of such micelles, a strategy was investigated for increasing the lipophilicity of the PPE block of these PEG-*b*-PPE amphiphilic copolymers. A PEG-*b*-PPE copolymer bearing pendant vinyl groups along the PPE block was synthesized and then modified by thiol-ene click reaction with thiols bearing either a long linear alkyl chain (dodecyl) or a tocopherol moiety. Ketoconazole was used as model for hydrophobic drugs. Comparison of the drug loading with PEG-*b*-PPE bearing shorter pendant groups is reported evidencing the key role of the structure of the pendant group on the PPE backbone. Finally, a first evidence of the biocompatibility of these novel PEG-*b*-PPE copolymers was achieved by performing cytotoxicity tests. The PEG-*b*-PPE derived by tocopherol was evidenced as particularly promising as delivery system of poorly water-soluble drugs.

## 1. Introduction

The ring-opening polymerization (ROP) of cyclic phosphoesters, especially phosphates, into biodegradable and biocompatible aliphatic polyphosphoesters is known since the 70s and the pioneering works of Penczek [1,2,3,4,5]. Since then, thanks to the development of the organo-catalysis, 5-membered cyclic phosphate monomers became very popular due to their fast and selective polymerization into high molar mass aliphatic polyphosphoesters (PPE) [6,7,8,9]. Thanks to the pentavalency of the phosphorus atom, a wide diversity of those cyclic monomers can be made available by esterification of 2-chloro-1,3,2-dioxaphospholane 2-oxide with various alcohols. Many alcohols being available, the pendant group of aliphatic PPEs can easily be modified, enabling the fine tailoring of a set of properties such as (bio)degradation rate, solubility in organic solvents and in water, hydrophilic/lipophilic balance and reactivity for post-functionalization. If these 5-membered phosphate rings polymerize efficiently by ROP, their high sensitivity towards hydrolysis makes their purification more delicate. Fractional distillation is efficient for enough volatile monomers. Nevertheless, this purification technique is no more valid for high molar mass monomers exhibiting a high boiling point. Therefore, a two-step approach emerged that consists in the ROP of a cyclic phosphate substituted by a functional group allowing the chemical post-modification of the PPE. Efficient reactions that occur in conditions mild enough to avoid the PPE degradation have to be selected. This is the case for reported examples that used the Cu-catalyzed azide-alkyne cycloaddition (CuAAC) [10,11,12], the thiol-ene [13,14,15,16,17,18] or the thiol-yne [19,20,21] reactions. In this work, we applied the thiol-ene two-step strategy to obtain PPE with long alkyl pendent chains with the ultimate goal to drastically increase the lipophilicity of PPE.

Indeed, this work pays attention to amphiphilic diblock copolymers made up of hydrophilic PEG and hydrophobic PPE blocks. These copolymers are prone to self-assemble into nanoparticles in water [22,23] which makes them very attractive for drug delivery applications. In a previous work, we demonstrated that increasing the length of the PPE substituent from ethyl to heptyl favorably impacts the size and stability of the nanoparticles [23]. In order to increase further the lipophilicity of the PPE aiming at improving the PPE affinity for lipophilic drugs, we investigated here the thiol-ene reaction for the synthesis of new PEG-*b*-PPE with highly lipophilic PPE block. For that purpose, PEG-*b*-PPE bearing butenyl side-groups (PEG-*b*-PBenEP) synthesized following an already reported procedure, ref. [23] has been used to investigate the thiol-ene addition of dodecane-1-thiol (dodec-SH) or a derivative of tocopherol substituted by a thiol function (Toco-SH) (Scheme 1). The choice of tocopherol, a liposoluble vitamin, is based on its use in the food and pharmaceutical industries, several drugs being ‘tocophilic’, i.e., exhibit a high affinity for tocopherol [24], so as for biomedical applications [25,26,27]. The self-assembly of these new PEG-*b*-PPE copolymers into nanoparticles was investigated by dynamic light scattering and pyrene fluorescent probe spectroscopy. Their potential for loading poorly water-soluble drugs was proved by selecting ketoconazole as a model drug. Finally, cytotoxicity tests by live/dead cell viability assays were performed for both types of copolymers as a first evaluation of their biocompatibility.

## 2. Results

### 2.1. Synthesis of Thiol Functional Tocopherol (TocoSH)

Owing the straightforward synthesis of well-defined PEG-*b*-PPE copolymer with pendent unsaturation on the PPE backbone, the thiol-ene reaction was selected to bring the lipophilic side-chains by post-functionalization of the PPE. Therefore, lipophilic thiol derivatives should first be made available. If dodecane-1-thiol (dodecSH) is a commercially available compound, it is not the case for thiol derivatives of tocopherol. Therefore, we developed an efficient strategy to introduce a thiol function onto tocopherol.

The esterification between α-tocopherol and thioglycolic acid, in presence of *N*,*N*′-dicyclohexylcarbodiimide (DCC) and 4-(dimethylamino)pyridine (DMAP) being not efficient (less than 10% of coupling), even for three days of reaction, we synthesized mercaptoethyl-d-α-tocopheryl succinate (TocoSH). For that purpose, d-α-tocopheryl succinate (Toco-succ) was reacted with a slight excess of 2-mercaptoethanol (1.2 eq.), in the presence of DCC (1.2 eq.) and a catalytic amount of DMAP (0.2 eq.) in CH_2_Cl_2_ heated at 35 °C for three days to form the ester linkage, as illustrated in Scheme 2.

After filtration to remove the dicyclohexylurea formed by the hydration of DCC, the organic solution was washed with different aqueous solutions: (i) HCl 0.05 M, (ii) saturated NaHCO_3_ and (iii) water and then dried on Na_2_SO_4_. The esterification reaction was confirmed by proton nuclear magnetic resonance (^1^H NMR) spectroscopy (Figure 1). The comparison of the protons of the -*O*-CH_2_-ester (peak K) to the signal of the -CH_2_- of the tocopherol part (peak M) allows the determination of the reaction yield, which reached 88%. With both thiols in hands, the post-polymerization thiol-ene reaction onto the PEG-*b*-PPE copolymers can be studied.

### 2.2. Synthesis of the Starting PEG-b-PBenEP Copolymer

The organocatalyzed homopolymerization of cyclic phosphate bearing various small-size (<C_10_) alkyl, alkene or alkyne pendant chains from a PEG macroinitiator being already well described, [18,23] we applied the previously reported conditions to synthesize the PEG-*b*-poly(2-but-3-enoxy-1,3,2-dioxaphospholane 2-oxide) (PEG-*b*-PBenEP) used as starting diblock copolymer. The ^1^H NMR spectrum and the size exclusion chromatography (SEC) trace of this copolymer are shown in Figure 2a and Figure 3a, respectively. They will be used as a reference for compa rison with the copolymers obtained after the click reaction with both thiols as depicted on Scheme 1.

### 2.3. Post-Polymerization Functionalization by Thiol-ene Reaction

The thiol-ene click reaction was carried out onto PEG-*b*-PBenEP under UV irradiation with an excess of the thiol compounds (1.5 eq. and 4.6 eq. of dodecSH and TocoSH, respectively and in the presence of a catalytic amount of 2,2-dimethoxy-2-phenylacetophenone (DMPA) as a photo-initiator, in similar conditions previously reported by Wooley et al. [28] A larger excess of Toco-SH allows to increase the number of tocopherol moieties grafted onto the butenyl groups of the PPE block. After reaction, the excess of the thiol compound was easily removed by the repeated precipitation (three times) from CH_2_Cl_2_ into cold diethyl ether, both thiol compounds are well soluble.

The comparison of the ^1^H NMR spectra before and after thiol-ene reaction with dodecane-1-thiol is illustrated in Figure 2a,b and shows the completeness of the functionalization by the total disappearance of the vinyl proton resonances at 5.8 and 5.20 ppm (labelled A and B in Figure 2a) in Figure 2b and the appearance of the proton resonance of the dodecane linear chains (between 1.72 and 0.92 ppm, labeled as I, J, K and L, respectively). In addition, the constant integration ratio (I_(C+D)_/I_E_) before and after the click reaction evidences the integrity of the PPE block during the post-polymerization modification. Moreover, the comparison of the integration of the peak L with the one of peaks (C+D) confirms the quantitativity of the thiol-ene click reaction.

When the thiol-ene reaction is performed with TocoSH on the PEG-*b*-PBenEP copolymer, the reaction is only partial as demonstrated by the ^1^H NMR spectra before and after the thiol-ene reaction (Figure 2c) where only a partial disappearance of the vinyl proton resonances (5.8 and 5.1 ppm, as labeled as A and B, respectively) is observed beside the appearance of the protons resonance of the tocopherol side-chain (between 2.9 and 0.8 ppm, labelled as G’, H, I and J). A conjugation efficiency of 60% was calculated by comparing the intensity of the protons B remaining vinyl groups and E of PEG used as internal reference and confirmed by the intensity ratio of the peaks E and J coming from the tocopherol. This conjugation yield is not improved by increasing the irradiation time, the bulkiness of the tocopherol might be the reason of the limited grafting reaction yield.

SEC analyses of the copolymers after thiol-ene reaction showed peak shifts to shorter elution time, relative to the block copolymers before the thiol-ene reaction, but no broadening of the elugram showing that the reaction does not degrade the copolymer, as illustrated in Figure 3. Indeed, similar bimodal SEC traces were observed for the starting PEG-*b*-PBenEP and both copolymers obtained after the thiol-ene reaction. It is noteworthy to recall that the small peak at low elution time corresponds to a second population of chains, attributed to the presence of a triblock copolymer (below 10%), PPE-PEG-PPE, due to some homotelechelic HO-PEO-OH in the commercial MeO-PEO-OH as evidenced by MALDI-TOF mass spectrometry analysis (Appendix A). Since the separation of this copolymer mixture is tedious, the starting amphiphilic copolymer was used without further purification for the thiol-ene reaction. The macromolecular characteristics of the amphiphilic copolymers obtained after the thiol-ene reaction are summarized in Table 1.

### 2.4. Thermal Characterization of the Copolymers

Since long alkyl chains were introduced in the copolymers as pendant groups of the PPE block, they could provide crystallinity to the polyphosphoester block of the materials. The DSC thermograms of the novel copolymers are shown in Figure 4. Figure 4a clearly evidences a melting temperature at −3 °C for the PEG-*b*-PdodecS-BEP, the main crystallization peak at 54 °C being due to the PEG block. Such derivatization with a dodecyl side group appears thus, to impart crystallinity to PPE. In contrast, we showed in a previous paper [23] that PPE bearing shorter alkyl side chains are all amorphous.

The DSC curve of the PEG-*b*-PTocoS-BEP copolymer shows the main melting peak at 52 °C corresponding to the crystallization of the PEG block and a slope change observed at −31.6 °C corresponding to the T_g_ of the PPE block. Even if this T_g_ value is still below 0 °C, it appears at a higher temperature than the one obtained for common PEG-*b*-PPE block copolymers (generally T_g_ < −50 °C) [23]. This also evidences the impact of the bulky pendent chains introduced along the backbone that tend to rigidify the PPE segments.

### 2.5. Behavior in Water of the PEG-b-PPE Diblock Copolymers

The self-assembly in water of the amphiphilic diblock copolymers before and after the thiol-ene reaction was first investigated by dynamic light scattering (DLS). The as-obtained size and the size distribution (PDI) of the self-assemblies in Milli-Q water are reported in Table 2.

In line with the previous studies [23], the copolymer with the pendant butenyl groups forms large and loose aggregates about 220 nm, similar to the corresponding copolymer bearing the saturated *n*-butyl groups (D_h_ = 126 nm, i.e., also above 100 nm). Remarkably, both novel amphiphilic copolymers self-assembled directly in water into small and well-defined micelles with a diameter of 22 nm and 85 nm respectively. A similar behavior was observed for PEG-*b*-PHEP, a copolymer with a *n*-heptyl pendant groups on the PPE block (D_h_ = 16 nm) [23]. The PDI remains above 0.3 for all the copolymers which could be due to the presence of the PPE-PEG-PPE triblock. A spherical morphology was observed by Transmission Electron Microscopy (TEM) for all the three copolymer micelles, as illustrated in Figure 5.

The CMC of the amphiphilic copolymers was determined by the pyrene fluorescence probe spectroscopy method and is reported in Table 2. Clearly, after the thiol-ene post-polymerization modification, the micellization occurred at a concentration two and three orders of magnitude lower owing to the increased hydrophobicity of the PPE segment. The very low CMCs, 1.1 × 10^−7^ and 8.3 × 10^−8^ moL/L respectively and the small size of the formed self-assemblies of the PEG-*b*-PDodecS-BEP and PEG-*b*-PTocoS-BEP amphiphilic copolymers make them both promising candidates for carriers of poorly water-soluble drugs.

### 2.6. Ketoconazole Encapsulation and Release

The key advantage of such polymer micelles is their ability to solubilize hydrophobic drugs into their core. Yang et al. [22] previously reported on the impact of increasing the alkyl chain length (from *n*-butyl to *n*-octyl) on the release profile of doxorubicin. Nevertheless, the report on the loading capacity (LC) of these different micellar systems is lacking. Hence, the loading capacity of a highly hydrophobic antifungal agent, ketoconazole (solubility in water 0.025 mmoL/L; log P = 4.3) [29], used as model drug, was investigated for the three PEG-*b*-PPE block copolymers. For that sake of comparison, the ketoconazole LC was also determined for amphiphilic PEG-*b*-PPE with the same polymerization degree for both blocks but exhibiting different pendant groups on the PPE block, i.e., PEG-*b*-PButEP (*n*-butyl as pendant groups), PEG-*b*-PHEP (*n*-heptyl as pendant groups). The ^1^H-and ^31^P NMR spectra (Appendix A) and SEC analysis (Appendix A) of these copolymers used in the present study are given as Appendix A.

The preparation of ketoconazole-loaded nanoparticles was conducted either by direct dissolution in water or by means of an organic co-solvent as more commonly reported to increase the LC [30]. Dichloromethane (CH_2_Cl_2_) was selected as co-solvent since it is a good solvent for both the copolymer and ketoconazole and because it has a low boiling point allowing its elimination by evaporation. Table 3 summarizes the LC in self-assembled micelles obtained via both processes for each copolymer system of decreasing HLB.

For all the systems, the use of the co-solvent for the micelles preparation enhanced the ketoconazole LC as compared to the direct dissolution in water. Indeed, due to the high hydrophobicity of ketoconazole, it has an extremely low solubility in water. When using an organic co-solvent, the LC is sensibly increased.

Comparing the copolymers bearing n-butyl or butenyl side-chains evidences that the presence of a vinyl function on the side chain has a marked influence on the LC of ketoconazole. While they have the same HLB, a higher loading is observed for butenyl side groups. Based on the structure of ketoconazole, π-π interactions between the aromatic rings and the vinyl group of the side-chains would favorably influence the drug encapsulation. The LC is also clearly improved when the length of the alkyl pendant group on the PPE block is increased whatever the micelles preparation processes (Table 3, top to bottom). Remarkably, the PEG-*b*-PTocoS-BEP encapsulates more ketoconazole than all the other tested PEG-PPEs, whatever the preparation process. The ketoconazole LC is increased by a factor 2 and 3 when 60% of the butenyl are derivatized by the tocopherol. That might be due to synergistic effects of (i) increasing the core hydrophobicity and (ii) providing unsaturations for π-π interactions with the ketoconazole. This clearly highlights that not only the increase of the HLB, by increasing the length of the pendant groups, favors the LC but also the possible specific interactions between the drug and the tailor-made side chains.

The ketoconazole release profiles from the various particles prepared by solvent evaporation are compared on Figure 6. In most cases, a fast release is observed whatever the copolymer, about 80% or more of the loaded content is released after 20 h. In contrast, a slower release without Burst effect is observed in the case of PEG-*b*-PTocoS-BEP micelles. Indeed, these micelles released only 9% during the first three hours. After these three first hours, the ketoconazole release rate is very similar to obtain a percentage of release of about 60% after 20 h and 75% after 48 h, i.e., significantly slower than the release rates recorded for the other PPE-based micelles.

### 2.7. In Vitro Cytotoxicity of PEG-b-PPE Copolymers

Cytotoxicity of the different copolymers to Human Umbilical Vein Endothelial Cells (HUVEC) was evaluated by live/dead cell viability assay. First, it was checked by ^1^H-NMR that no signal from residual amount of polymerization catalysts and thiol-ene photocatalyst, i.e., DBU/TU and DMPA, is detected. Then, solutions of increasing concentrations of the different PEG-*b*-PPE amphiphilic block copolymers were prepared in the endothelium basal medium (EBM) cell culture. In that EBM medium, fluorescence spectroscopy in presence of pyrene evidences the presence of the copolymers as unimers for the lowest used concentration and confirms the presence of the micelles for all the higher concentrations reported on Figure 7.

Cell viability after 24 h incubation (Figure 7) revealed that when copolymers are present in the medium as unimers, i.e., at the lowest concentration used, they do not show significant cytotoxicity for HUVEC whatever the side-chain. In contrast, the cytotoxicity of the micelles depends on the nature of the PPE pendant group. PEG-*b*-PBenEP showed a lower cytotoxicity after 24 h of incubation for 5.5 g/L copolymer concentration than PEG-*b*-PButEP (Figure 7a,b). Their pendant groups containing 4 carbon atoms only differ by the presence of the unsaturation which appears to decrease the cytotoxicity. Nevertheless, at higher concentrations, both systems become cytotoxic. A comparable evolution is found for PEG-*b*-PHEP, i.e., increase of the cytotoxicity related to the increase of the concentration. Nevertheless, the micelles of the latter being smaller (D_h_ = 16 nm), they appear less cytotoxic (some living cells are kept at 33.5 g/L).

In case of the copolymers obtained by thiol-ene reaction, the cytotoxicity tests evidence a clear impact of the thiol on the HUVEC viability. In line with the copolymers described above, the PEG-*b*-PDodecS-BEP exhibit increasing cytotoxicity with concentration (Figure 7d).

Quite remarkably, the cell viability assessed after 24 h of incubation with PEG-*b*-PTocoS-BEP at different concentrations (Figure 7e) revealed no cytotoxicity for HUVEC. Indeed, the percentages of viable cells of 100% are observed for all solution concentrations up to 25 g/L. By comparison with the cytotoxicity results obtained for all the other PEG-*b*-PPE, the tocopherol moieties provide exceptionally high biocompatible carrier as demonstrated by the absence of cytotoxicity of these self-assembled micelles for HUVEC.

## 3. Materials and Methods

### 3.1. Materials

Methanol (MeOH, Acros), diethyl ether (Chem-lab), 2,2-dimethoxy-2-phenylacetophenone (DMPA, Aldrich, St. Louis, MO, USA), dodecane-1-thiol (Aldrich), d-α-Tocopheryl succinate (Toco-Succ, Aldrich), 2-mercaptoethanol (Aldrich), *N*,*N*′-dicyclohexylcarbodiimide (DCC, Aldrich), 4-(dimethylamino)pyridine (DMAP, Aldrich) and calcium hydride (CaH_2_, Aldrich) were used as received without further purification. 1,8-diazabicyclo [5.4.0]undec-7-ene (DBU, Aldrich) were dried over CaH_2_ at room temperature and purified by distillation under reduced pressure just before use. Monomethoxy poly(ethylene oxide) (MeO-PEG-OH, Aldrich) was dried by three azeotropic distillations with anhydrous toluene before use. Toluene (Chem-lab), tetrahydrofuran (THF, Chem-lab) and dichloromethane (CH_2_Cl_2_, Chem-lab) were dried on molecular sieves. Milli-Q water was used for the preparation of micelles. 1-[3,5-bis(trifluoromethyl)phenyl]-3-cyclohexyl-2-thiourea (TU), 2-butoxy-1,3,2-dioxaphospholane 2-oxide (ButEP), 2-heptyloxy-1,3,2-dioxaphospholane 2-oxide (HEP) and 2-butenoxy-1,3,2-dioxaphospholane 2-oxide (BenEP) monomers and their copolymers with PEG were synthesized as described elsewhere in the literature. [23,31] Ethidium homodimer-1 (EthD-1) and Hoechst 33,324 were purchased from Thermo Scientific. Human Umbilical Vein Endothelial Cells (HUVEC-2) and Cell culture medium EBM-PRF were acquired from BD Biosciences and Lonza, respectively.

### 3.2. Instruments

^1^H and ^31^P NMR analyses were performed on a Bruker Advance 250 MHz spectrometer in CDCl_3_ at 25 °C in the FT mode. The ^31^P NMR measurement was carried out with the ^1^H powergate decoupling method using a 30° flip angle. ^1^H NMR chemical shifts are reported in ppm relative to Me_4_Si. ^31^P NMR chemical shifts are reported in ppm relative to H_3_PO_4_. Size exclusion chromatography (SEC) was carried out in THF at 45 °C at a flow rate of 0.7 and 1 mL/min with Viscotek 305 TDA liquid chromatograph equipped with 2 PSS SDV analytical linear M columns calibrated with polystyrene standards. The Differential Scanning Calorimetry (DSC) was performed using a DSC Q500 (TA instruments) calibrated with indium. The sample is introduced in the calorimeter at room temperature and is cooled down to −80 °C. A first temperature ramp (10 °C min^−1^) is applied up to 100 °C to eliminate the polymer’s thermal history. Then, the sample is again cooled until −80 °C and heated with a temperature ramp (20 °C min^−1^) up to 100 °C. The melting temperature (T_m_) is recorded during the second heating scan.

### 3.3. Synthesis

Synthesis of mercaptoethyl-d-α-tocopheryl succinate (TocoSH). In a 250 mL round-bottom flask equipped with a magnetic stirring bar, DCC (2.33 g, 11.3 mmol), DMAP (230 mg, 1.9 mmol) and d-α-Tocopheryl succinate (Toco-Succ) (5 g, 9.4 mmol) were added and dissolved in CH_2_Cl_2_ (50 mL). After stirring at room temperature for 10 min, 2-mercaptoethanol (900 mg, 11.5 mmol) was added. The mixture was heated at 35 °C for 3 days away from the light. After filtration, the solution was washed three times with an aqueous solution of hydrochloric acid ([HCl] = 0.05 M), three times with a saturated aqueous solution of NaHCO_3_ and finally, three times with water. After drying over Na_2_SO_4_, the organic phase was filtered and the solvent was removed under reduced pressure. 5 g of a pale-yellow oil was recovered.

Thiol-ene modification of the PEG-*b*-PBenEP copolymer with dodecyl thiol. 3 g of PEG-*b*-PBenEP (0.42 mmol, M_n_ = 6700 g/mol) and 56 mg of DMPA (0.22 mmol) were dissolved in 5 mL of anhydrous CH_2_Cl_2_. 1.3 g of 1-dodecanethiol (0.64 mmol) was added to the solution and the reaction solution was stirred under UV radiation (365 nm, 200W) at room temperature. After 45 min, the copolymer was recovered by precipitation in cold diethyl ether three times and filtrated before being dried under vacuum.

Thiol-ene modification of the PEG-*b*-PBenEP copolymer with Toco-SH. 2.5 g of PEG-*b*-PBenEP (1.40 mmol, M_n_ = 6900 g/mol) and 54 mg of DMPA (0.22 mmol) were dissolved in 5 mL of anhydrous CH_2_Cl_2_. A total of 4 g of Toco-SH (6.4 mmol) was added to the solution and the reaction solution was stirred under UV radiation (365 nm, 200 W) at room temperature. After 1 h, the polymer was recovered by precipitation in cold diethyl ether three times and filtrated before being dried under vacuum.

### 3.4. Micelles Size and Morphology

Preparation of self-assembled micelles. Typically, 50 mg of the PEG-*b*-PPE copolymer were placed in 20 mL of milli-Q water and stirred for 2 h.

Size and morphology measurements. Particle size and size distribution were acquired from freshly prepared micelles solutions (filtered with a microfilter having an averaged pore size of 0.45 µm) by employing dynamic light scattering (DLS). Measurements were carried out in a glass cells at 25 °C at a measuring angle of 165° and repeated five times in order to check their reproducibility. The morphology of the micelles was investigated by Transmission Electron Microscopy (TEM). A small drop of each aqueous copolymer solution after DLS analysis was deposited onto a formvar coated copper grid. The excess of the copolymer solution was wiped off using filter paper and the grid was let dried and stored under ambient atmosphere until analysis with a Philips CM100 microscope equipped with an Olympus camera and operated by a Megaview system equipped computer.

Determination of critical micelles concentration (CMC). CMC values in Milli-Q water for the different amphiphilic block copolymers were determined by the pyrene probe fluorescence spectroscopy using a LS50B luminescence spectrometer (Perkin Elmer). Typically, solid pyrene (2.6 × 10^−8^ mol) was dissolved in aqueous solutions of the amphiphilic copolymer of increasing concentration from 1 × 10^−11^ to 1 × 10^−3^ mol/L. Milli-Q water was used to dissolve first the copolymer before adding it to vials containing the pyrene. Since the amount of needed pyrene is quite low (5.2 µg), a solution of pyrene (1.3 × 10^−3^ moL/L) in acetone was first prepared. A total of 20 µL of the pyrene solution was transferred into glass vials and acetone was evaporated at room temperature before adding the copolymer solution. After one night of stirring at room temperature, the fluorescence spectra of pyrene were recorded from 360 to 460 nm after an excitation at 335 nm. The emission and excitation slit widths were set at 3.0 and 3.5, respectively. The ratio of the peak intensities at 373 nm and 383 nm ($I_{373}/I_{383}$) of the emission spectra were calculated for each copolymers solution and recorded as a function of the copolymer concentration.

### 3.5. Ketoconazole Encapsulation

Encapsulation by direct dissolution. A total of 50 mg of the PEG-*b*-PPE copolymer and 5 mg of ketoconazole were directly dissolved in 20 mL of Milli-Q water under stirring at room temperature for 2 h.

Encapsulation by solvent evaporation. A total of 50 mg of the PEG-*b*-PPE copolymer and 5 mg of ketoconazole were completely dissolved in 1 mL of CH_2_Cl_2_ before adding 20 mL of Milli-Q water. The solution was stirred at 25 °C for 2 h to evaporate of CH_2_Cl_2_.

Quantification by HPLC analysis. At the end of the encapsulation procedures described above, only a part of the ketoconazole was encapsulated in the micelles. The excess precipitated in the nanoparticle solution. This free solid ketoconazole was thus filtered through a 1.2 µm filter. The filtered solution was immediately diluted in acetonitrile (micelles solution/acetonitrile: 1/1) to quantify the encapsulated ketoconazole. This quantification was performed by a validated high performance liquid chromatography (HPLC) by using a reversed-phase HPLC (Agilent 1100 series, Agilent Technologies). The samples were analyzed at 220 nm in 80% methanol/20% ammonium acetate solution (0.5%) using a 125/4 Nucleodur RP C18 column 5 µm at 25 °C. A calibration curve was constructed using different concentrations of free ketoconazole (0.01–2 mg/mL). The drug loading capacity (LC, %, *w*/*w*) of the micelles was calculated with the following Equation (1):(1)$$\mathrm{LC}\;\left(\%,w/w\right)=\frac{\mathrm{Amount}\;\mathrm{of}\;\mathrm{ketoconazole}\;\mathrm{incoporated}\;\mathrm{in}\;\mathrm{micelles}\;\left(\mathrm{mg}\right)}{\mathrm{Amount}\;\mathrm{of}\;\mathrm{copolymer}\;\left(\mathrm{mg}\right)}\times 100$$

Ketoconazole release profiles. Ketoconazole-loaded micelles solutions (20 mL) free from the excess of non-encapsulated drug were prepared at a polymer concentration of 2.5 mg/mL from each PEG-*b*-PPE block copolymers. The loaded micelles solutions were transferred into dialysis bags (MW cut-off: 3500 Da, supplied by Spectrum Laboratories) and placed into PBS solution. At selected time intervals, 500 µL of micelles solution was withdrawn from inside the dialysis bag and diluted in acetonitrile to quantify the decreasing ketoconazole amount remaining in the micelles by HPLC.

Live/Dead cell viability assay. Cytotoxicity of polymer micelles was evaluated by determining the viability of HUVEC cells after incubation in EBM-PRF with different concentrations of the amphiphilic copolymer depending on their CMC. Live/dead cell viability assay was performed using BD Falcon 96-wells HTS Imaging microplates containing cells and different copolymer solutions. After 24h incubation (at 37 °C and 5% of carbon dioxide), Hoechst 33,342 (800 × 10^−9^ M) and EtHD-1 (4 × 10^−6^ M) were added to distinguish between viable and non-viable adherent cells. Cells were imaged automatically with a BD Pathway 855 high content analyzer (BD Biosciences). BD Attovision software (BD Biosciences, version 1.6) was used for both image acquisition and individual cell segmentation. Flow cytometry software (Kaluza 1.2; Beckman Coulter, Mijdrecht) was used to analyze numerical data on total cell numbers and dead cells. Number of viable cells was calculated via the following equation: number of total adherent cells (Hoechst positive nuclei) minus number of dead cells (EtHD-1 positive nuclei) per well and averaged for replicates (*n* = 3–4 per copolymer solutions). Data were plotted as average viable cells number for the different amphiphilic copolymers at the different copolymer concentrations with untreated cells as control set at 100%. Student t-test was applied to define statistical differences, with *p*-value below 0.05 considered significantly different (GraphPad Prism 5.01 for Windows, GraphPad Software, San Diego, CA, USA).

## 4. Conclusions

PEG-*b*-PDodecS-BEP and PEG-*b*-PTocoS-BEP amphiphilic block copolymers were obtained through the chemical modification of the pendant vinyl functions of the preformed PEG-*b*-PBenEP copolymer via a very efficient thiol-ene reaction. These amphiphilic block copolymers self-assembled directly in water into well-defined small and spherical micelles. Moreover, the CMC values measured for these two novel amphiphilic copolymers are very low, suggesting that they will remain stable upon dilution in the body fluids. These self-assembled nanoparticles are thus particularly well-suited to encapsulate poorly soluble drug, as demonstrated here with ketoconazole. Indeed, ketoconazole was successfully encapsulated during the micelle formation, which significantly enhanced the ketoconazole concentration in aqueous medium. The tocopherol-containing micelles give the highest ketoconazole loading content. Burst release is not observed with these micelles that exhibit slower release as compared to other systems.

The in vitro evaluation of the cytotoxicity of these novel copolymers demonstrated that no toxicity for HUVEC is revealed when the cells are incubated with PEG-*b*-PTocoS-BEP nanoparticles, whatever the copolymer concentration. In conclusion, this tocopherol modified polyphosphoester appear as safe and most promising material for controlled drug delivery systems of poorly water-soluble drugs.

## Data Availability

Not applicable.

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
