# Peer review of "Thiol-ene Reaction: An Efficient Tool to Design Lipophilic Polyphosphoesters for Drug Delivery Systems"

_molecules, 2021, doi:10.3390/molecules26061750_

Round 1

Reviewer 1 Report

Jérôme and coworkers reported the synthesis of poly(phosphoesters) and its functionalization via thiol-ene reaction. Several issues are suggested to be addressed. 

  1. The structure of PEG-b-PTocoS-BEP in Scheme 1 should be revised. The current version indicates that the unconjugated and conjugated alkene units belong to the same block, meaning that there is a fixed orientation. In fact, the thiol-ene reaction on PEG-b-PBenZP is expected to result in the unreacted alkene and Toco-conjugation statistically distributed. Please revise.
  2. Line 164 cited a reference to prove the impurity within the starting material. It is more convincing to run a GPC for the starting material (MeO-PEO-OH) and show this evidence in the supporting information.
  3. In Table 1, does longer irradiation time result in higher yield for the toco-conjugation, e.g. 2 hours?
  4. Characterization (NMR, GPC, etc.) for PEG-b-PButEP, PEG-b-PBenEP, are PEG-b-PHEP is missing. 
  5. The unit for polymer concentration is suggested to be revised to mg/mL or μg/mL (IUPAC suggestion: https://iupac.org/cms/wp-content/uploads/2016/01/Compendium-of-Polymer-Terminology-and-Nomenclature-IUPAC-Recommendations-2008.pdf). Figure 6 can indeed benefit from such revisions. 
  6.  What is the drug release mechanism and drug release kinetics of these polymeric micelles? Please provide the data. 

Reviewer 2 Report

This is a well-written manuscript about the preparation of a novel diblock copolymer-based drug delivery system. 

The only question of the reviewer is related to the bimodal SEC traces of the copolymers. According to the authors, the small peak at low elution volumes corresponds to the presence of PPE-PEG-PPE triblock copolymers. Can the authors estimate the amount of this fraction in the preparations? What happens with the triblock copolymer when the samples are dispersed in water? Do these polymers form co-micelles or separate micelles? Can this heterogeneity be observed on DLS or TEM?

Minor points: Please check/correct the sentences at lines 39 and 77. 
